# Importance of Metalloproteinase Enzyme Group in Selected Skeletal System Diseases

**DOI:** 10.3390/ijms242417139

**Published:** 2023-12-05

**Authors:** Monika Kulesza, Aleksandra Kicman, Joanna Motyka, Tomasz Guszczyn, Sławomir Ławicki

**Affiliations:** 1Department of Population Medicine and Lifestyle Diseases Prevention, Medical University of Bialystok, 15269 Bialystok, Poland; monika.kulesza@sd.umb.edu.pl (M.K.); motyka.k.joanna@gmail.com (J.M.); 2Department of Aesthetic Medicine, Medical University of Bialystok, 15267 Bialystok, Poland; olakicman@gmail.com; 3Department of Pediatric Orthopaedics and Traumatology, Medical University of Bialystok, 15274 Bialystok, Poland; tomasz.guszczyn@icloud.com

**Keywords:** metalloproteinases, skeletal system diseases, bone tumors, osteoporosis, intervertebral disc degeneration

## Abstract

Bone tissue is a dynamic structure that is involved in maintaining the homeostasis of the body due to its multidirectional functions, such as its protective, endocrine, or immunological role. Specialized cells and the extracellular matrix (ECM) are responsible for the remodeling of specific bone structures, which alters the biomechanical properties of the tissue. Imbalances in bone-forming elements lead to the formation and progression of bone diseases. The most important family of enzymes responsible for bone ECM remodeling are matrix metalloproteinases (MMPs)—enzymes physiologically present in the body’s tissues and cells. The activity of MMPs is maintained in a state of balance; disruption of their activity is associated with the progression of many groups of diseases, including those of the skeletal system. This review summarizes the current understanding of the role of MMPs in bone physiology and the pathophysiology of bone tissue and describes their role in specific skeletal disorders. Additionally, this work collects data on the potential of MMPs as bio-markers for specific skeletal diseases.

## 1. Introduction

Bone is a form of mineralized and vascularized connective tissue that performs a variety of functions in the human body. Such functions include locomotion, protection (the protection of internal organs and bone marrow), and storage (the storage of calcium and phosphate), among others. It is also recognized as an immune and endocrine organ [1,2]. The most important bone-building cells include osteoblasts, osteocytes, and osteoclasts—these cells act antagonistically or synergistically to each other. The bone extracellular matrix, also known as osteoid, is a product of osteoblast activity [1,3,4].

The ECM is responsible for the remodeling of specific bone structures. The major components of the ECM are fiber-forming proteins such as collagen, elastin, fibronectin, glycoproteins (tenascin, vitronectin, entactin), lamina, proteoglycans (syndecan-1, aggrecan), polysaccharides (hyaluronic acid), and glycosaminoglycans (GAGs). In addition, the ECM is made up of fibrils that form type-I collagen and type-II collagen, found mainly in cartilage [5,6]. One of the elements responsible for ECM homeostasis comprises matrix metalloproteinases (MMPs), which are a family of zinc-dependent proteolytic enzymes involved in the degradation of ECM [7,8]. MMPs are involved in a number of physiological phenomena in the body, but their excessive activity is associated with the development of many diseases [5,9,10].

The exact role of MMPs in the human skeletal system is not very well studied; most reports focus on animal models. These enzymes are involved in physiological processes related to bone homeostasis; however, they have also been shown to be involved in the formation of a number of bone disorders, such as osteoporosis or osteoarthritis [11,12,13]. A particularly important role of these enzymes has also been shown in bone malignancies such as osteosarcoma or Erwing’s sarcoma [14,15].

Bone diseases are still a diagnostic challenge because the number of bone-related markers is low. The medical field is constantly evolving to discover markers with a greater sensitivity and specificity for the structure under study. MMPs are involved in bone-related inflammatory processes; therefore, the purpose of this work was to summarize the existing knowledge of the role of matrix metalloproteinases in the skeletal system, to determine their importance in selected bone diseases, and to establish their potential as possible biomarkers in bone diseases.

## 2. Bone Tissue Overview

### 2.1. Cells of Bone Tissue

Each bone cell type (osteoblast, osteocyte, or osteoclast) has a specific function, and they participate synergistically or antagonistically with each other during bone matrix formation or degradation. Osteocytes make up 90–95% of the population and play an important role in regulating bone remodeling and sensing mechanical activation [16,17,18,19].

Osteoblasts are bone-forming cells and account for 4–6% of the cells in bone. These cells produce bone matrix, as well as coordinating matrix mineralization and offering an endocrine function. Osteoblasts synthesize many organic molecules, such as type-I collagen or osteocalcin, and inorganic molecules, such as proteoglycans, and secrete MMPs [16,17,19,20].

Osteoclasts make up 1–2% of bone cells. During bone resorption, osteoclasts secrete cytokines, hormones, enzymes (i.e., cathepsin K), and growth factors (i.e., transforming growth factor-β (TGF-β)), which are stored in the bone matrix. Osteoclasts resorb the mineralized matrix and promote the remodeling of the organic fraction of bone, while osteoblasts are responsible for its formation via depositing specialized components of the ECM before its proper mineralization [3,16,17,19,21]. The balance between osteoblasts and osteoclasts is crucial for the maintenance of appropriate bone mass, and an imbalance in this synchronization contributes to the development of bone disease [1,2,22].

### 2.2. Bone Extracellular Matrix

The organic ECM of bone (also called osteoid) is mainly synthesized by osteoblasts before the occurrence of mineralization. The ECM of bone consists of three fractions: organic (proteins: 15–20% and lipids: 3%), inorganic (65–70%), and collagen-bound water (10%). This composition varies with age, sex, bone location, and health status. The organic components of ECM can be divided into collagen and non-collagen proteins. Collagen proteins are the main structural biopolymer and the main determinant of the mechanical properties of connective tissues. In bone, they are the most common component of the organic ECM and consist only of fibrillar types (types I, III, and V). Type-I collagen accounts for about 90% of the collagen in bone and is organized at several hierarchical levels (molecular level, collagen fibrils, and collagen fibrils) to provide bone with its mechanical properties [17,23].

### 2.3. Bone Remodeling

Maintaining a balance between bone formation and bone resorption is fundamental to the proper functioning of the skeletal system. The regulation of calcium levels, and the processes of regenerating bone damage and bone turnover depend on this balance. The loss of this balance, on the other hand, will result in a reduction or excessive increase in bone mass, leading to the development of various disease disorders [17].

The process of bone remodeling is a conjugated process between the resorption and reconstruction of the bone matrix, and the full model of reconstruction is complex and still not fully understood. Thus, in order to maintain an appropriate balance between these processes, the interactions between osteocytes, osteoclasts, and osteoblasts must be maintained under strict control, which involves different signaling molecules. For the moment, it has been established that one of these signaling axes is the receptor activator of the nuclear factor-κB (RANK)/RANK ligand (RANKL)/osteoprotegerin (OPG) pathway [24], but the importance of other molecules, such as MMPs, that are involved in the control of cell–cell interactions is further being investigated.

The primary microenvironment of bone remodeling is basic multicellular units (BMUs), which consist of various types of cells, mainly osteoclasts and osteoblasts. In the remodeling bone area, where osteocytes have undergone apoptosis, local BMUs are recruited to rebuild bone [25]. Increased apoptosis is primarily observed in areas where microdamage to bone has occurred due to passive use or damage from trauma [17]. They can also be caused by chronic inflammation, hormone deficiency—particularly estrogen deficiency—genetic factors, or glucocorticoid intake [17,26,27].

## 3. The Role of Metalloproteinases in Physiology and Pathology, with Special Emphasis on Bone Tissue

MMPs are a family of enzymes whose activity is dependent on zinc ions [9,11,16]. At least 25 different MMPs have been characterized so far in vertebrates [28]. MMPs in the human body are encoded by 24 genes, including two identical genes on chromosome 1 encoding MMP-23: MMP23A and MMP23B [28]. MMPs are produced by cells such as fibroblasts, endothelial cells, mast cells, microglia cells, odontoblasts, dendritic cells, smooth muscle myocytes, and keranocytes [11]. The expression of MMPs is also found in bone cells such as osteoblasts [29,30,31,32] and osteoclasts [33]. In the case of osteocytes, the expression of single MMPs (MMP-2 and MMP-8) has been confirmed in rat cells [34]. Importantly, chondrocyte-like cells building the nucleus pulposus and the inner fibrous ring of intervertebral discs have been demonstrated to show immunopositivity for MMP-1, MMP-2, MMP-3, and MMP-9, among others [12,35]. The expression of MMPs has been shown to change with age—infants and children showed no evidence of these enzymes, while a group of adolescents and adults revealed a significant level of cellular expression of MMP-1, MMP-2, and MMP-3, while MMP-9 expression was low [35]. This indicates that MMPs are involved in the regulation of intervertebral disc homeostasis [12].

Although teeth have a different structure than bones, due to their close association with the mandible and maxilla, they are included in the skeletal system. The secretion of particular MMPs is found in the cells that build the dental apparatus, such as ameloblasts during enamel development, dentin odontoblast cells, and predentin [36,37,38,39,40,41,42,43].

Based on their substrate specificity, metalloproteinases can be divided into collagenases, gelatinases, stromelysins, matrilysins, membrane-type MMPs (MTs), and others [11]. The classification of MMPs is shown in Figure 1.

MMPs play an important role in many biological processes, including mediating cell-to-cell adhesion, tissue remodeling, cell migration, invasion, proliferation, and apoptosis [11]. They can cleave growth-factor-binding proteins or latent growth factors, thereby regulating their synthesis and release into the extracellular environment [44,45]. In addition, MMPs are thought to play an important role in regulating the viability and function of osteoclasts, osteoblasts, and osteocytes, and chondrocyte proliferation and differentiation [8]. In the early stages of bone development, MMP-9 seems to have the most important role; presumably, this enzyme mediates bone implantation and resorption [46]. MMP-16, produced by osteoblasts and osteocytes, is also responsible for bone development. It degrades ECM proteins, such as type-I collagen, thereby promoting bone growth and development [46]. MMP-2 and MMP-3 have also been shown to mediate the maintenance of bone homeostasis. MMP-2 is responsible for bone mineralization, as its reduced activity translates into the decreased mineralization of this tissue and generalized osteolysis [47]. In the case of MMP-13, it has been shown to be involved in intrachondral ossification and cartilage degradation. In addition, a reduced expression of MMP-13 has been implicated in slowing osteoclastic bone resorption [48].

MMPs are also involved in all phases of tooth development, mainly through mediating the processes of proliferation and apoptosis, as well as the degradation and mineralization of tooth tissue. A particularly important role is demonstrated by metalloproteinase 20 (MMP-20), which is involved in the development of enamel [39,43,49]. Mutations in the gene encoding MMP-20 have been proven to cause disorders in proper enamel development, such as amelogenesis imperfecta. The enamel of patients with this disorder is characterized by a soft and rough surface with foci of pigmentation. In addition, patients are at risk for more rapid cavity formation and earlier tooth loss [37,50,51,52]. In addition, some studies indicate that these enzymes are implicated in tooth eruption [49].

Beyond physiological phenomena, MMPs are associated with the initiation and progression of diseases in many systems, such as the cardiovascular, nervous, and excretory systems. The particular importance of these enzymes has been confirmed in the course of cancer [5,10,11]. Some studies also indicate an important role of MMPs in the course of skeletal diseases, which will be presented in later chapters in this article.

The activity of MMPs is controlled via tissue inhibitors of metalloproteinases (TIMPs). These compounds covalently bind to the corresponding MMPs and, thereby, inhibit their proteolytic activity. Currently, four different molecules from the group of tissue inhibitors of metalloproteinases—TIMP-1, TIMP-2, TIMP-3, and TIMP-4—have been recognized. These compounds, in addition to inhibiting the action of MMPs, also participate in physiological phenomena in the body, and the disruption of the equilibrium in the MMP–TIMP system leads to the progression of many pathological conditions [1,5,9,10,11]

## 4. The Role of MMPs in Selected Skeletal Diseases

### 4.1. Degenerative Spine Disease

Degenerative spine disease is a progressive and chronic disease associated with the premature degeneration of the tissues that connect the elements of the spine. Degenerative spine disease currently has an incompletely understood etiology but is characterized by a decrease in cell numbers, the appearance of inflammation, and a loss of extracellular matrix [53,54]. Tissue degeneration results from repeated micro- and macro-trauma, metabolic processes, and overlapping risk factors, such as age, gender, type of work performed, and genetic factors [55]. Depending on the location, cervical, thoracic, and lumbar spinal degeneration are distinguished. A sizable number of studies indicate the involvement of MMPs in the degeneration of spinal tissues—most current reports focus on intervertebral disc degeneration (IVD). Patients diagnosed with IVD are noted to have an elevated expression of MMPs such as MMP-1, MMP-3, MMP-8, MMP-9, MMP-10, MMP-12, MMP-13, and MMP-14 [12,53,56,57,58,59]. Also, the execution of immunohistochemical staining has shown increased levels or activity of particular enzymes from this group: MMP-1, MMP-3, and MMP-13 [12,35,60].

The expression of MMPs has been correlated with the severity of the disease or its histopathological features. As reported by Aripaka et al. [53], a higher expression of MMP-1, MMP-3, MMP-10, and MMP-13 was observed in vertebral samples from patients with more advanced IVD expressed in Pfirrmann grades. The same correlation was also found by Rutges et al. [61] for MMP-14 protein expression and the degree of disc degeneration. There was no relation between Pfirrmann’s degree and MMP-2 expression. This disagrees with the work of Crean et al. [58], who found an increase in the expression levels of both MMP-2 and pro-MMP-2 dependent on the degree of vertebral degeneration. The same study showed an equal relationship for MMP-9. Studies on the expression of MMPs in IVD patients were also conducted by Bachmeier et al. [56], who proved that the strongest increase in the expression in degenerated discs was found for MMP-3. This enzyme also showed the strongest positive correlation with histopathological changes. Also, for the expression of MMP-8, a positive correlation has been found with the presence of histopathological changes and the duration of pain. In contrast, such correlations are not found for MMP-2, MMP-9, MMP-1, and MM-13. These studies partially agree with the work of Weiler et al. [35], who also showed a correlation between the expression of individual MMPs and the occurrence of histopathological changes. There was a positive correlation between the expression of MMP-1, MMP-2, and MMP-3 and the appearance of clefts and tears in the nucleus pulposus, and between the expression of MMP-1 and MMP-2 and the appearance of cracks in the fibrous ring. A correlation was also found between the levels of MMP-1, MMP-3, and MMP-9 and the appearance of histological changes such as changes in cell density, the appearance of granular lesions, and myxoid degeneration. It should be noted, however, that in the case of MMP-9, such a correlation was found only for the fibrous ring. Other studies on MMP-9 expression have been performed in patients with lumbar disc degeneration, and it was shown that in the group with more advanced vertebral lesions, MMP-9 expression was higher than in patients with milder forms of the disease. Studies of MMP-9 expression were supported by the performance of immunohistochemical assays that indicated an equal relationship [59]. A single study also indicates an important role for MMP-12 in spinal degeneration. A higher expression of this enzyme was observed in degenerated discs compared to healthy tissues and was additionally associated with an increased expression of fibrosis markers such as α-SMA, FSP1, and FAP-α [57]. Some studies have been conducted using immunohistochemical staining—a study by Le Maitre et al. [12] showed a correlation between an increase in the number of cells positive for MMP-1, MMP-3, and MMP-13 and the degree of disc degeneration according to the Pfirrmann scale. Also, Deng et al. [62] and Hingert et al. [63] confirms the importance of MMP-1 in IVD—MMP-1 levels were higher in degenerated discs compared to healthy tissues. Moreover, patients in the highest, stage-III (severe) disease had a higher percentage of staining for MMP-1 compared to patients in stages II (moderate) and I (mild). A single study also indicates that the ratio of MMP-1 to IHC (immunohistochemical expression) is an independent predictor of the severity of cervical or lumbar disc degeneration [64].

Interestingly, one study demonstrated the potential of MMP-1 as a prognostic serum marker: patients in stage III showed higher levels of this enzyme compared to stages I and II [62]. This may indicate the preliminary usefulness of MMP-1 as a potential prognostic marker in intervertebral disc (IVD) degeneration.

### 4.2. Malignant Diseases of Bone and Dental Cancers

#### 4.2.1. Osteosarcoma

Osteosarcoma (OS) is among the most common malignant bone tumors in children and adolescents [65,66,67]. The etiology of this disease is unclear, and lesions are usually located in the distal part of the femur [65,67]. Radiological studies are necessary to establish a diagnosis. In the case of laboratory tests, most patients have increased levels of alkaline phosphatase (ALP) and lactate dehydrogenase; however, 30–40% of patients show no changes in biochemical parameters [65,68].

Part of the research has focused on the role of MMPs in OS patients as potential tumor markers. The expression of pro-MMP-2 [69], MMP-2 [14,70,71,72,73], MMP-8 [14], MMP-9 [59,74,75,76,77], MMP-13 [14], MMP-14 [70], and MMP-26 [14] were confirmed in OS samples. Interestingly, in the case of MMP-2, MMP-9, MMP-13, and MMP-26, immunohistochemical staining showed their presence in both the primary and metastatic foci of OS lung metastasis. In the case of MMP-8, expression was limited only to the primary focus of the tumor [14], while the mRNA levels for MMP-2 and MMP-9 did not differ between the primary and metastatic foci [70]. However, the mRNA levels for MMP-3 and MMP-26 were not compared. In addition, in the case of MMP-2, it was shown that this enzyme was partially coexpressed with CXCR4 [72], while high levels of the MMP-9 protein were positively correlated with high levels of pre-ALP (alkaline pre-phosphatase) in the serum of OS patients [74]. Studies on the relationship between MMPs expression in OS patients and prognosis are often contradictory. As reported by Gong et al. [72], high MMP-2 expression was associated with the presence of distant metastatic OS and abbreviated rates of overall survival and metastasis-free survival rates. In another study, MMP-2 expression was associated with the presence of lung metastases [73]. However, according to the work of Korpi et al. [14], MMP-2 expression levels were not associated with changes in overall survival, while, as reported by Zhang and Zhang [73], RT-qPCR studies showed that levels of this enzyme did not depend on location and tumor stage (according to Enneking’s stage). In the case of MMP-9, most reports showed that a high expression of this enzyme was associated with increased OS mortality [59], shorter overall survival, and the presence of metastatic foci compared to patients with a low or undetectable MMP-9 expression [75,77]. In addition, according to Vaezi et al. [15], a higher MMP-9 expression was characteristic of highly malignant, metastatic, and recurrent tumors; these parameters also contribute to the unfavorable prognosis of patients with OS. However, a single study indicates that MMP-9 expression is not dependent on the mean tumor microvascular density or any clinical parameters of the lesion [76]; nevertheless, in our opinion, MMP-9 shows preliminary potential as a marker of poor prognosis in OS patients.

The activity of MMP-2 and MMP-9, which was confirmed via zymography [69,78], was also studied in OS patients. According to the work of Kunz et al. [78], MMP-2 and MMP-9 were the predominant enzymes showing activity in patients’ biopsy specimens. The activities of these enzymes differed between good and poor responders to treatment. MMP-9 activity was high in patients with a good response to chemotherapy, while high MMP-2 activity was associated with a poor response to treatment. Importantly, for MMP-2 expression studies, Korpi et al. [14] shows a similar relationship—patients with a good response to chemotherapy showed low MMP-2 expression.

Some research has also been conducted using serum from OS patients. According to Kushlinskii et al. [79], MMP-2 levels in cancer patients were lower than in healthy people. This disagrees with the work of Kushlinsky et al. [80], who found no difference in MMP-2 levels in OS patients compared to healthy people. An identical relationship was also found for MMP-7 [80]. The same study also found lower levels of MMP-9 in cancer patients, while MMP-9 levels were independent of gender, age, location, and bone tumor size [80]. The acquisition of lower concentrations of MMP-2 and MMP-9 in patients with this bone cancer is surprising; however, it may indicate the potential of these compounds in the diagnosis and prognosis of OS patients. This requires continued research; however, it should be noted that the initial potential of these compounds as tumor markers has been demonstrated in the course of other conditions, such as ovarian [81], breast [82], and colorectal [83] cancers.

#### 4.2.2. Ewing’s Sarcoma

The second-most-common bone cancer in children and adolescents is Ewing’s sarcoma (ES), a disease most often associated with a fusion transcript involving the EWS-FLI1 or EWS-ERG genes. This type of cancer is characterized by an extremely unfavorable course and high metastatic potential [66]. As in the case of OS, some data indicate an association of ES with enzymes from the metalloproteinases group. The expression of individual MMPs is also found in the tissues of patients with ES. Tissue analysis revealed the presence of MMP-2, MMP-9, and MMP-14 [15,84,85,86,87]. MMP-1 and MMP-3 are not found [86,88], and the highest levels of expression have been demonstrated for MMP-9 and MMP-14 [86]. Additionally, in the case of MMP-14, its expression in both primary and metastatic foci has been confirmed [87].

Few data are available on the relationship between MMP-9 and MMP-14 expression levels and patient prognosis. According to Brookes et al. [84], a high expression of MMP-14, demonstrated via immunohistochemistry, was associated with reduced event-free and overall survival in patients with ES. On the other hand, as Vaezi et al. [15] reported, high mRNA expression for MMP-9 positively correlated with tumor size, higher malignancy grade, chemotherapy status, and a tendency toward recurrence and metastasis. Also, MMP-9 protein levels correlated with tumor malignancy grade, chemotherapy status, and recurrence tendency [15]. It is unfortunate that there are currently no studies on the use of MMPs as plasma or serum markers in ES.

#### 4.2.3. Chondrosarcoma

Other malignant primary bone tumors include chondrosarcoma; this tumor, unlike ES, is more common in adults but, like ES, is of mesenchymal origin. The prognosis of chondrosarcoma varies and depends mainly on the stage of the tumor and its subtype.

Similar to OS and ES, the expression of MMPs such as MMP-1 [65,89,90,91,92], MMP-2 [93,94,95], MMP-3 [89,95], MMP-7 [95,96], MMP-9 [89,94,95], and MMP-14 [93] is also found in biopsy specimens from patients with chondrosarcoma. In the case of studies and MMP-2 and MMP-14, the expression of this enzyme was confirmed in various molecular subtypes of chondrosarcoma: clear-cell chondrosarcomas, mesenchymal chondrosarcomas, conventional chondrosarcomas, and dedifferentiated chondrosarcomas [93]. MMP-8 expression was studied in a single study that indicated that only some of the tumors examined (5 of 28) showed MMP-8 expression at a level capable of detection via PCR [90]. The expression of MMP-13 in patients with chondrosarcoma is a matter of controversy. As reported by Malcherczyk et al. [89], immunohistochemical studies did not reveal the presence of MMP-13 in biopsy material. This contradicts the studies of Yao et al. [92], Sugita et al. [95], and Uria et al. [97], who demonstrated, using the same method as Malcherczyk et al. [89], the expression of MMP-13 in chondrosarcoma. Importantly, a single study also confirms mRNA expression for MMP-13 in patients with this type of cancer [90].

Importantly, some studies also indicate the occurrence of MMPs such as MMP-1, MMP-2, MMP-13, and MMP-14 in benign bone tumors—enchondromas [92,93]. However, it is a matter of dispute whether the expression of these enzymes differs between benign and malignant lesions. This is important because the determination of enzymes from this group could potentially serve as an adjunctive test in differentiating malignant from benign lesions; such potential has been tentatively demonstrated, for example, in ovarian cancer [81]. The expression of MMP-2 and MMP-14 was not significantly different between enchondroma and chondrosarcoma [93]. In contrast, according to a study by Yao et al. [92], the expression of MMP-1 and MMP-13 is higher in chondrosarcoma than enchondroma.

Similar to OS and ES, some studies have indicated associations between MMPs expression and the prognosis of patients with chondrosarcoma. However, it should be emphasized that these data are often mutually exclusive, pointing to the need for further studies to clearly define the role of MMPs as predictive or prognostic factors in bone tumors.

In the case of MMP-2, most reports suggest that a higher expression of this enzyme was observed in chondrosarcoma of a higher histologic grade and in recurrent tumors [94,95]. Like with MMP-1, studies have not shown a relationship between MMP-2 expression and patient prognosis [95]. Interestingly, a high expression of MMP-9 appears to have a protective effect. High levels of MMP-9 expression were associated with a better histological differentiation of chondrosarcoma—more differentiated tumor type (grades I and II) showed higher MMP-9 expression than grade III chondrosarcoma [89]. This is not consistent with the work of Sugita et al. [95], who found no correlation between MMP-9 expression and the histological grade of the neoplasm. According to Malcherczyk et al. [89], a high expression was also associated with metastatic potential in the disease and prolonged overall survival.

Malcherczyk et al. [89] found no correlation between histologic grade, metastatic potential, and time versus the expression of overall-survival MMP-1. Interestingly, some studies indicate the prognostic potential of the MMP-1/TIMP-1 ratio—patients with recurrent chondrosarcoma have shown higher values of the MMP-1/TIMP-1 ratio than patients without recurrence [90,91].

The relationship between MMP-13 expression and histologic grade is a contentious issue. According to Malcherczyk et al. [89], there is no correlation between MMP-13 expression and the grade of chondrosarcoma. However, as reported by Sugita et al. [95], chondorsarcoma of a higher grade is characterized by higher MMP-13 expression. At the same time, no correlation has been shown between MMP-13 expression and patient prognosis [90,95]. In addition, the MMP-13/TIMP-1 ratio does not correlate with patient survival [90]. Also, MMP-8 expression does not correlate with patient survival and prognosis [90]. For MMP-3, which belongs to the stromelysin group, the same results were obtained as for MMP-13 [89,90,95].

#### 4.2.4. Dentigerous Tumors

Several scientific data also relate to the importance of MMPs in tumors of dental origin. The expression of particular MMPs is found in various types of dental tumors, such as ameloblastoma, ameloblastic carcinoma, adenomatoid odontogenic tumors, calcifying cystic odontogenic tumors, and odontoma [98,99,100,101,102]. In patients with calcifying cystic odontogenic tumors, the presence of MMPs has been associated with the progression of this type of tumor [99].

### 4.3. Osteoporosis and Osteopenia

Osteopenia and osteoporosis are metabolic diseases in which resorption processes are increased, resulting in decreased bone density and the formation of pathological fractures. The onset of osteoporosis and osteopenia is compounded by a number of interrelated factors, such as endocrine disruption (especially menopause), age, and comorbidities [103,104]. As specific MMPs are involved in bone turnover, research is being conducted among these compounds to further our knowledge of their role in these diseases.

Increased levels of MMP-2 [105,106,107,108,109], MMP-9 [110,111,112], and MMP-13 [13,113] have been observed among osteoporotic patients compared to those with normal BMD levels, among both men and women. In 2003, the Mansell et al. team [114] was the only one to examine MMPs levels in the active enzyme and proenzyme forms in osteoporotic bone tissue and compare the results to a healthy bone sample. According to this study, the level of both forms of MMP-2 in osteoporotic tissue was higher than in healthy bone; however, statistical significance was not achieved in the study. It is worth noting that the study groups were very small (12 test tissues and 7 control tissues), which may have been the direct reason why the threshold for statistical significance was not exceeded. MMP-2 in the serum of patients with osteoporosis was also significantly elevated compared to patients with osteopenia [106,107,108]. In turn, the MMP-9 gene was singled out during gene clustering analysis as one of the secondary genes in terms of involvement in osteoporosis [115]. Circulating serum MMP-9 showed a trend of increasing levels from patients with a normal BMD, through patients with a reduced BMD, to patients with established osteoporosis [110,112]. Only one study of all available human studies showed no significant differences in MMP-2 and MMP-9 levels in patients with osteopenia and osteoporosis compared to healthy controls among older patients [116]. MMP-13, on the other hand, showed increased serum levels in the osteopenia patient group compared to healthy controls, while the concentrations were equalized relative to the osteoporosis group [13].

Large discrepancies in the information concern the correlation of MMP-2 levels with markers of bone turnover. In patients with osteoporosis, MMP-2 levels correlated negatively with BMD [106,107,108,109] and correlated positively with levels of osteocalcin (OC), serum cross-linked N-telopeptides of type I collagen (NTX), bone alkaline phosphatase (BALP) [106,107,108], tartrate-resistant acid phosphatase 5b (TRACP-5b) [117], or other factors, such as sB7-H3 protein [105]. In contrast, Thisiadou et al. [118] found an inverse correlation between MMP-2 and OC, as described in the years 2005–2006, while there was no correlation between MMP2 and N-terminal propeptide of type-I collagen (PINP) and serum beta-CrossLaps (β-CTX). In contrast, Dai et al. [109] observed no association between MMP-2 and BALP levels, and Xiao et al. [117] correlated them negatively with BALP and vitamin D receptor (VDR) levels.

Many studies focus on the topic of osteoporosis as a secondary disease often coexisting with other chronic clinical syndromes, such as chronic obstructive pulmonary disease (COPD) and idiopathic pulmonary artery hypertension (IPAH), evaluating different age groups. However, all of these papers note one seemingly invariable theme—a negative correlation between serum MMP-9 levels among patients and BMD index [110,111,112,119,120]. Additionally, a positive correlation between MMP-9 and other markers of bone turnover, such as the RANKL/OPG ratio [112] and β-CTX [120], has also been noted. In their study, however, Bolton et al. [110] compare healthy patients with a normal BMD to COPD patients with a normal BMD, finding no differences in MMP-9 levels. In addition, they examine the ability of MMP-9 in classifying patients with respect to the onset of osteoporosis. The power of such an assay assessed using the ROC curve among COPD patients was AUC = 0.86, while it was as high as 0.84 among all study participants [110]. On the other hand, in a study by Zhang et al. [112], they showed a weak positive correlation between MMP-9 and tumor necrosis factor α (TNF-α). The results from both teams suggest that MMP-9 levels will depend only slightly on chronic inflammation and more on processes related to bone turnover. In addition, in women with current osteoporosis, MMP-9 levels decreased significantly after the introduction of bone-protective factors into their lifestyle in the form of regular 3-month balanced training [121] or therapeutic treatment with three different compounds: alendronate, risedronate, and ralocifene [122]. However, this theory needs more research.

Both studies on MMP-13 were conducted on rather small groups, with about 30-50 subjects in each subgroup, which makes the interpretation of the results difficult. Nevertheless, both studies undeniably outlined the relationship between serum MMP-13 levels and bone metabolic status. The divergent information regarding the correlation is concentrated within mineral density. According to the study by Dai et al., MMP-13 was negatively correlated with BMD levels [13] not only in the group of patients with osteoporosis, but also in the group of patients with osteopenia, while Zhu et al. [113] observed no such correlation in the group of women with osteoporosis. In addition, MMP-13 was negatively correlated with BALP [113], estradiol (E2), and OPGL levels [13], and positively correlated with OPG, PINP [13], and runt-related transcription factor 2 (Runx2) [113] among patients with osteoporosis. In a group of women with osteopenia, MMP-13 again negatively correlated with E2 levels, as well as with CTX levels [13] Interestingly, a 6-month therapeutic course with recombinant human parathormone among women with osteoporosis clearly reduced serum MMP-13 levels by almost 37%. Unfortunately, the authors did not relate the post-therapy MMP-13 levels to MMP-13 levels among healthy women [13].

Lv et al. [123] conducted an analysis on genetically predicted levels of serum MMPs and concluded that for the European population, MMP-1, MMP-3, MMP-7, MMP-8, MMP-10, and MMP-12 showed no evidence of association with BMD index assessed via DEXA. These predictions appear to be confirmed by results obtained by some research teams around the world. Guo et al., Luo et al., and Zhang et al. [106,107,108], examining levels of serum MMP-1, and Farhan et al., examining MMP-12 [124], found no differences between the levels of these MMPs among women with osteoporosis and osteopenia and healthy individuals. Additionally, correlating MMP-1 concentrations with markers of bone turnover, they noted the lack of correlation of MMP-1 with such markers as BMD, OC, NTX, and BALP. In contradiction are the results published by Liang et al. [125], which show higher MMP-1 concentrations among men with osteoporosis compared to healthy controls. However, they did not evaluate the correlation between MMP-1 concentrations and calcium concentrations, or BMD index, in their work. In contrast, the correlation between MMP-1 and markers of bone turnover was examined by Thisiadou et al. [118], who studied bone mineral status among patients with chronic kidney disease. According to their data, MMP-1 levels showed a positive correlation with OC, PINP, and β-CTX.

In the case of MMP-3, no correlation was observed either between serum levels and the incidence of osteoporosis [126] or BMD index [127]. However, among patients with autoinflammatory and autoimmune diseases, such as systematic lupus erythematosus (SLE), rheumathoidal arthritis (RA), or ankylosing spodylitis, MMP-3 seemed to gain diagnostic significance in assessing bone status. In RA patients in the group with osteoporosis and osteopenia, MMP-3 was significantly elevated in serum compared to RA patients with a normal BMD [47]. In addition, MMP3 levels were higher in RA patients with osteoporosis than among patients with osteopenia, but the authors did not assess the level of statistical significance for these groups [47]. Among patients with SLE, serum MMP-3 was negatively correlated only with the BMD of the lumbar spine, but not of the femoral neck [128], while, among patients with osteoporosis secondary to ankylosing spodylitis, it was correlated positively with TRACP-5b and negatively with BALP and VDR [117]. Studying only postmenopausal women, work by Dai [129] and Abdu Allah et al. [130] also noted significantly higher levels of serum MMP-3 among women with developed osteoporosis than women with a normal BMD. However, the discrepancy between the results of Dai and of Ablu Allah et al. relates to the group of women with osteopenia. According to the work of Dai [129], the serum MMP-3 in women with osteopenia did not differ in level from women with a normal BMD, but was significantly lower than in women with osteoporosis. In contrast, Abdu Allah et al. [130] observed that the MMP-3 in women with osteopenia was higher than in women with a normal BMD, with no differences from women with osteoporosis. In the group of women with osteoporosis, MMP-3 levels correlated negatively with BMD and OPG ligand (OPGL), and positively with OPG [129] and osteopontin (OPN) [130]. In contrast, in a group of women with osteopenia, MMP-3 correlated negatively only with the BMD of the lumbar spine and ward angle [129]. It was also hypothesized that MMP-3 interacting with OPN may be an initiating factor in the process of osteoporosis at postmenopausal age. In addition, MMP-3 was examined as a diagnostic factor using the ROC curve, obtaining results for the sensitivity, specificity, and power of the test at a very high level of 93%, 84%, and AUC = 0.9520, respectively [130].

There are no studies on MMP-8 levels among patients with osteoporosis. However, there is a paper evaluating serum MMP-8 levels in relation to skeletal characteristics among children and adolescents with obesity. In this study, neither among a control group of children with normal BMI nor among children with obesity was there any correlation between MMP-8 and bone turnover markers such as BALP, CTXm and PINP [131], which would agree with the predictions of Lv et al. [123].

### 4.4. Metastatic Bone Disease

Bone and bone marrow are among the most common sites for cancer metastasis [132]. Bone tissue is highly vascularized and has niches containing trophic and growth factors, which enable efficient tumor colonization in the bone matrix [133]. Bone metastases, in many cases, are asymptomatic and are detected incidentally during follow-up examinations for other diseases [134]. The prevention and therapy of bone metastases is a priority in the treatment of cancer patients [132].

Metastatic bone disease (MBS) is a common complication of cancer, and about 70% of cancers metastasizing to bone originate from advanced lung, breast, kidney, thyroid, or prostate cancer [132,133,134,135,136]. Bone metastases can occur as lytic, blastic, or mixed lesions [134]. Because bone metastases are mostly untreatable, they are associated with high patient mortality [133]. A number of studies have suggested that metalloproteinase enzymes may be involved in the formation of bone metastases [137]. Ongoing studies on this group of enzymes suggest that cancer cells can secrete and/or induce osteoclasts to release enzymes, including MMP-2 and MMP-9, into the bone microenvironment [138,139].

The study by Incorvaia et al. found higher levels of MMP-2 and MMP-9 in breast and prostate cancer patients compared to controls. These studies may indicate a potential role for these enzymes in cancer progression; on the other hand, the Incorvaia et al. study showed no significant differences as to the levels of the molecules tested in cancer patients with metastatic bone disease compared to breast/prostate cancer patients without MBD. Similar research results were obtained by Skerenova et al. [140]; MMP-2 levels were significantly elevated in breast cancer patients with bone metastasis. The study by Nemeth et al. [141], similarly to the study by Incorvaia et al., showed that MMP-2 and MMP-9 are specifically associated with the metastasis of prostate cancer. In addition, the plasma and urine levels of these enzymes were correlated with prostate cancer metastasis [141]. MMP-9 levels were positively correlated with PSA levels in prostate cancer patients, which may indicate that this metalloproteinase is involved in bone metastasis [138]. MMP-9 could play an important role in prostate cancer metastasis to bone through two mechanisms: the first mechanism acts on ECM and bone cell activity, while the second has a potential direct effect on the cancer cells themselves [142]. There are studies confirming the correlation between MMP-9 expression and disease progression, where MMP-9 levels are significantly higher in advanced stages of the disease, thus promoting bone metastasis [142].

The study by Pivetta et al. suggests that MMP-13 plays an important role in the microenvironment of bone metastasis in breast cancer patients [143]. MMP-13 is secreted by breast cancer cells following stimulation by osteoblasts or inflammatory mediators, including IL-8 [143].

## 5. Further Research Directions—Sterile Bone Necrosis

In addition to the disease entities described in this paper, skeletal disorders include other conditions, such as sterile bone necrosis. Sterile bone necroses are a group of diseases characterized by the necrosis of bone tissue without microbial involvement. The main cause of sterile bone necrosis is considered to be a disturbance in the blood supply to the tissue, secondary to a variety of etiologic factors such as trauma, certain medications (corticosteroids), the presence of other sterile bone necrosis in the patient, alcohol, or hematologic diseases such as sickle cell anemia [144,145,146]. Based on epidemiological data, sterile bone necrosis occurs mainly in children and adolescents who are active in sports, mainly due to the occurrence of repetitive trauma, which, as mentioned, plays a role in the pathogenesis of these diseases. The most common sterile bone necroses include Osgood–Schlatter disease, Kohler disease, Legg–Calvé–Perthes disease, and Blount disease. [144,145]

Since the clinical diagnosis of the disease is based on imaging studies, biochemical diagnosis of these diseases is currently underutilized, which also translates into a lack of reports on the role of MMPs in this group of diseases. However, based on the reports collected in this paper, it is postulated that these enzymes could be used as prognostic or predictive markers in sterile bone necrosis. The implementation of such markers would translate into better targeting of the therapeutic process and, as a result, could be associated with shorter hospitalization times for patients. Determining the potential of MMPs in sterile bone necrosis requires research; however, based on the information gathered, they can be considered as molecules with high prognostic and predictive potential.

## 6. Summary

Bone is a tissue that undergoes constant and dynamic remodeling; in this process, matrix metalloproteinases are involved. These enzymes are expressed in bone-forming cells and structures. Their activity is essential for the normal development of these structures; however, MMPs are also associated with the formation of bone diseases. At least 25 different MMPs have been identified in the human body, but the activity of these compounds varies depending on the type of bone disease and also correlates differently with the clinical features of the condition. Some studies also indicate that MMPs could be used in the future as potential progressive and predictive markers, but a clear determination of their potential requires further research. In order to better understand the content of the paper, a summary of the role and diagnostic potential of MMPs in selected bone diseases is summarized in Table 1.

## Figures and Tables

**Figure 1 ijms-24-17139-f001:**
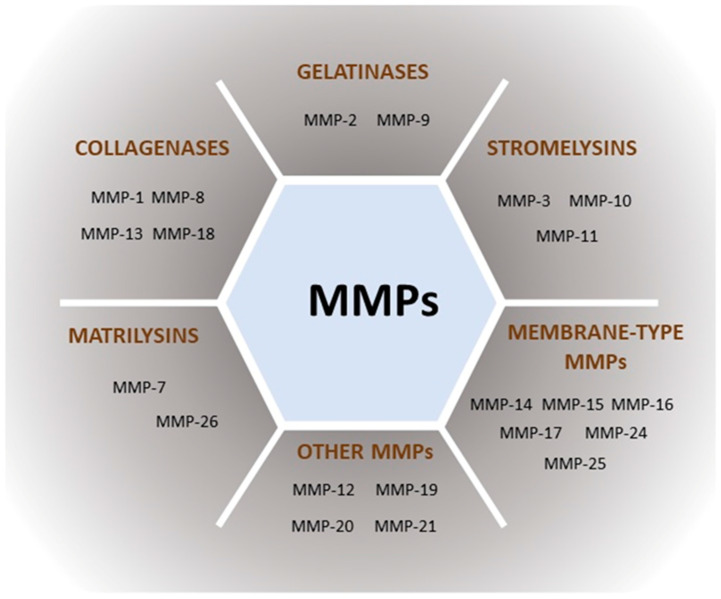
The classification of MMPs based on their substrate specificity.

**Table 1 ijms-24-17139-t001:** Summary of the role and diagnostic potential of MMPs in selected bone diseases.

INTERVERTEBRAL DISC DEGENERATION
*MMP-2*
-Higher expression in patients with more advanced IVD [58]. -Positive correlation between expression and the occurrence of clefts and tears in the nucleus pulposus and fractures in the fibrous ring [35]. -Positive correlation between the number of cells immunopositive for MMP-1 and the degree of disc degeneration [12].
*MMP-9*
-Higher mRNA and protein expression in IVD patients [58,59,60]. -Higher expression in patients with more advanced IVD [58]. -Positive correlation between expression and the presence of histological changes in the fibrous ring [35].
*MMP-1*
-Higher mRNA or protein expression and activity in patients with IVD [12,53] -Higher mRNA or protein expression in patients with more advanced IVD [12,53,62,63] -Positive correlation between expression and the occurrence of clefts and tears in the nucleus pulposus, ruptures in the fibrous ring, and the appearance of histological changes in the vertebra [35]. -Serum levels of MMP-1 higher in patients with more advanced IVD [62].
*MMP-8*
-Higher mRNA expression in IVD patients [56]. -Positive correlation between mRNA expression and the presence of histological changes [56]. -Positive correlation between mRNA expression and duration of pain [56].
*MMP-13*
-Higher mRNA or protein expression and activity in IVD patients [12,53]. -Positive correlation between the number of cells immunopositive for MMP-13 and the degree of disc degeneration [12]. -Higher mRNA expression in patients with more advanced IVD [53].
*MMP-3*
-Higher mRNA or protein expression and activity in IVD patients [12,35,53,60]. -The strongest increase in expression among all MMPs in degenerated discs [56]. -Strongest positive correlation of expression with the presence of histopathologic changes [56]. -Positive correlation between expression and the occurrence of clefts and tears in the nucleus pulposus, and the appearance of histological changes in the vertebra [35]. -Positive correlation between the number of cells immunopositive for MMP-3 and the degree of disc degeneration [12].
*MMP-10*
-Higher mRNA or protein expression in patients with IVD [53]. -Higher mRNA or protein expression in patients with more advanced IVD [53].
*MMP-12*
-Higher mRNA expression in IVD patients [57]. -Higher mRNA expression in degenerated discs compared to healthy tissues [57]. -Higher expression correlated with expression of fibrosis markers (α-SMA, FSP1, and FAP-α) [57].
*MMP-14*
-Higher protein expression in patients with IVD [61]. -Higher protein expression in patients with more advanced IVD [61].
BONE CANCERS
*OSTEOSARCOMA*
*MMP-2*
-Confirmed expression in OS samples [14,70,71,72,73] and in OS lung metastases [14]. -Confirmed enzymatic activity in OS samples [69,78]. -High activity [78] and high expression [14] correlated with poor response to treatment. -Simultaneous expression of MMP-2 with CXCR4 [72]. -High expression correlated with the presence of distant metastases, shortened OS and MFS rates [72], and the presence of lung metastases [73]. -Higher MMP-2 concentrations in OS patients compared to healthy subjects [79].
*MMP-9*
-Confirmed expression in OS samples [74,75,76,77] and in lung metastases [14]. -Confirmed enzymatic activity in OS samples [69,78]. -High activity correlated with good response to treatment [78]. -Positive correlation of MMP-9 protein expression with serum pre-ALP levels [74]. -High expression found in highly malignant tumors with a tendency toward recurrence and metastasis [15]. -High expression correlated with increased OS mortality [59], shorter OS, presence of metastatic foci [75,77]. -Lower serum concentrations in OS patients compared to healthy patients [80].
*MMP-8*
-Confirmed expression in OS samples [14]. -Expression restricted to primary focus only, no expression in metastatic foci [14].
*MMP-13*
-Confirmed expression in OS samples [14]. -Expression found in primary and metastatic foci [14].
*MMP-26*
-Confirmed expression in OS samples [14]. -Expression found in primary and metastatic foci [14]
*MMP-14*
-Confirmed expression in OS samples [14].
EWING SARCOMA
*MMP-2*
-Confirmed expression in ES samples [85,86,88].
*MMP-9*
-Confirmed expression in ES samples [15,85,86,88]. -Highest expression levels in ES samples [86]. -High mRNA expression correlated with tumor size, high degree of malignancy, chemotherapy status, tendency to recur, and metastasis [15]. -High protein expression correlated with high tumor malignancy, chemotherapy status, and tendency to recur [15].
*MMP-14*
-Confirmed expression in ES samples [84,86,87]. -Highest level of expression in ES samples [86]. -Expression found in primary and metastatic foci [87]. -High expression correlated with decreased event-free and overall survival [84].
*CHONDROSARCOMA*
*MMP-2*
-Confirmed expression in *chondrosarcoma* samples [93,94,95]. -Expression confirmed in different types of *chondrosarcoma—clear-cell chondrosarcomas, mesenchymal chondrosarcomas, conventional chondrosarcomas,* and *dedifferentiated chondrosarcomas* [93]. -High expression correlated with higher histologic grade and a tendency toward recurrence [94,95].
*MMP-9*
-Confirmed expression in *chondrosarcoma* samples [89,94,95]. -High expression correlated with better histological differentiation of the tumor [89]. -High expression correlated with prolonged OS in patients [89].
*MMP-1*
-Confirmed expression in *chondrosarcoma* samples [65,89,90,91,92]. -Expression higher in chondrosarcoma compared to benign lesions [92].
*MMP-8*
-Expression low, undetectable in some tumors [90].
*MMP-13*
-Confirmed expression present in *chondrosarcoma* samples [92,95,97]. -Expression higher in *chondrosarcoma* compared to benign lesions [92]. -Higher MMP-13 expression correlates with higher histological grade [95].
*MMP-3*
-Confirmed expression in *chondrosarcoma* samples [89,95].
*MMP-7*
-Confirmed expression in *chondrosarcoma* samples [95,96].
*MMP-14*
-Confirmed expression in *chondrosarcoma* samples [93]. -Confirmed expression in various types of *chondrosarcoma*—*clear-cell chondrosarcomas, mesenchymal chondrosarcomas, conventional chondrosarcomas*, and *dedifferentiated chondrosarcomas* [93].
ODONTOGENIC TUMORS
Few data on MMP expression and patient prognosis depending on MMP expression.
*MMP-2*
-Higher among osteoporotic patients than patients with a normal BMD [105,106,107,108,109] and osteopenia patients [106,107,108]. -Confirmed higher protein level in osteoporotic bone tissue in contrast to healthy bone [114]. -Contradictory results regarding the relationship between MMP-2 and bone turnover markers: (a) positive correlation with OC, NTX, BALP [106,107,108], TRACP-5b [117], and sB7-H3 [105], (b) negative correlation with BMD [106,107,108,109], OC [118], BALP, and VDR [117].
*MMP-9*
-Confirmed higher among osteoporotic patients than patients with a normal BMD [110,111,112] and osteopenia patients [110,112], -Selected as gene with secondary involvement in osteoporosis [115]. -Among patients with osteoporosis as secondary disease serum MMP-9: (a) negatively correlated with BMD index [110,111,112,119,120], (b) positively correlated with RANKL/OPG ratio [112], β-CTX [120], TNF-α [112], -The level of circulating MMP-9 decreases after osteoporosis management via training or treatment [121,122]. -The level of circulating MMP-9 has shown a high power of the test (AUC = 0.8400) as an osteoporosis blood marker [110].
*MMP-1*
-Confirmed higher concentration among patients with osteoporosis than among healthy patients [125]. -Positive correlation with OC, PINP, and β-CTX [118].
*MMP-8*
-Among children and adolescents, MMP-8 serum level did not correlate with BMI, nor BALP, CTX, or PINP [131].
*MMP-13*
-Confirmed higher MMP-13 serum level: (a) among osteoporotic patients than among patients with a normal BMD [13], (b) among osteopenia patients than among patients with a normal BMD [13]. -Confirmed negative correlation between MMP-13 serum level and BMD index in osteoporotic and osteopenia patients [13]. -In osteoporotic patients’ serum, MMP-13 was: (a) negatively correlated with BALP [113], E2 and OPGL [13], (b) positively correlated with OPG, PINP [13], and Runx2 [113], -In osteopenia patients’ serum, MMP-13 was negatively correlated with E2 and CTX [13] -The level of circulating MMP-13 decreased after osteoporosis management through treatment [13].
*MMP-3*
-Conflicting data among patients with osteoporosis as the sole disease: (a) genetically predicted serum level is not associated with BMD [123], (b) in postmenopausal women, a higher serum MMP-3 in osteoporotic women than in women with a normal BMD [114,130], (c) in postmenopausal women with osteopenia, the MMP-3 serum level is higher [130] or shows no difference [114] than in women with a normal BMD, (d) in postmenopausal women with osteopenia, the MMP-3 serum level is lower [114] or shows no difference [130] than in osteoporotic women. -Serum MMP-3 among only postmenopausal women: (a) with osteoporosis, it was correlated negatively with BMD and OPGL [114] and positively with OPG [114] and OPN [130], (b) with osteopenia, it was correlated negatively with the BMD of the lumbar spine and ward angle [114]. -Among patients with autoinflammatory and autoimmune diseases as their primary disease, the MMP-3 serum level was: (a) significantly higher in osteoporotic RA patients than in RA patients with a normal BMD [47], (b) higher (but not significantly) in osteoporotic RA patients than in RA osteopenia patients [47], (c) negatively correlated with the BMD of only the lumbar spine among patients with SLE [128], (d) positively correlated with TRACP-5b, and negatively with BALP and VDR, among patients with ankylosing spondylitis [117]. -MMP-3 is hypothesized to be an initiating factor with OPN of postmenopausal osteoporosis [130]. -In postmenopausal women, MMP-3 was presented as a good candidate for a blood marker for osteoporosis, with high sensitivity (93%), specificity (84%), and power of the test (AUC = 0.9520) [130].
*MMP-10*
-Genetically predicted serum level is not associated with BMD [123].
*MMP-7*
-Genetically predicted serum level is not associated with BMD [123].
*MMP-12*
-Genetically predicted serum level is not associated with BMD [123]. -It has been confirmed that there are no differences in MMP-12 concentration among osteoporosis, osteopenia, and normal-BMD patients [124].
STERILE BONE NECROSIS
-No data are available on the role of MMPs in these diseases. -MMPs may be considered in the future as potential candidates for prognostic and predictive markers.

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
