# Peer review of "Importance of Metalloproteinase Enzyme Group in Selected Skeletal System Diseases"

_ijms, 2023, doi:10.3390/ijms242417139_

Round 1

Reviewer 1 Report

Comments and Suggestions for Authors

General comments

 The  review of Kulesza et al. discuss the recent advances on the the role of matrix metalloproteinases in  normal and pathological bone remodeling and  their potential clinical usefulness as  biomarkers of  some specific bone disorders.

Although the topic appears timely, the revision process of the manuscript has highlighted  some critical issues  and several flaws  that the authors need to be addressed. In particular:

1.     In the Abstract,  the Authors  claim that the  purpose of this review is to provide an overview on  the role of MMPs in bone physiology and pathophysiology and  in specific skeletal disorders. (lines 22-24.) On the other hand, in the Introduction,  the Authors  assert  that the aim of the review is  to provide insight into the most recent advances on the role of MMPs in the skeletal system and  in the most common bone diseases, and to evaluate  their  clinical use  as possible biomarkers of these pathological conditions. Apart from  these  divergent  statements,  because the aim   reported  in the Abstract  appears to be consistent with the topic of the review, I would ask the authors to explain the reason why they have taken into consideration just some specific  pathological conditions.  Given the  epidemiologic,  and socio-economic burden  of  many other most commom malignant and non-malignant bone disorders it should be more fruitfull to provide insight into the most recent advances on the role of MMPs/TIMPs system and related microRNAs also in   non-malignant skeletal disorders such as primary and secondary osteoporosis, osteoarthritis (OA) , rheumatoid arthritis (RA), Paget’s disease of bone (PDB), osteomyelitis and  cancer related bone disease  such as Multiple Myeloma (MM) and Metastastic Bone Disease (MBD). In this setting, emerging evidence highlights the fact that, in MM and MBD the MPPS/TIMPs/microRNA system appear to play a pivotal role in promoting these pathological conditions and are actually considered potential therapeutic targets and useful biomarkers that may foster the development of nover and more effective clinical treatment of these diseases.

2. Regarding the role of MMPs in bone physiology and pathology, to be thorough a section describing the role of specific inhibitors of these enzymes, namely tissue inhibitors of metalloproteinases (TIMPs) in normal and pathological bone remodeling should be added. As the activity of MMPs  is  also regulated by TIMPs,  a physiological bone remodeling process  requires a  correct balance between MMPs and TIMPs. On the other hand, a growing number of preclinical and clinical studies has provided evidence that  Imbalanced MMPs/TIMPs activity underlies major bone pathologies. Furthermore, TIMPs act as signaling molecules endowed with cytokine-like activities, thus influencing, besides normal and pathological bone turnover, various biological processes, including cell growth, apoptosis, differentiation, angiogenesis, and oncogenesis.

Major Points

1.Introduction

Line 45 “….The exact role of MMPs in bone is not very well studied”…..To date, the involvement of MMPs in bone remodeling has been better defined following the experimental  use of MMP-knockout mouse models. These studies, by highlighting  a variety of bone abnormalities, have  provided clear evidence on the modulting  role of MMPs  in multiple stages of physiological and pathological bone remodeling processes (See for instance Hardy E et al.. Front Physiol. 2020 Feb 5;11:47. doi: 10.3389/fphys.2020.00047; Khoswanto C. J Oral Biol Craniofac Res. 2023 Sep-Oct;13(5):539-543. doi: 10.1016/j.jobcr.2023.06.002.  Liang HPO, (2016) Metalloproteinases in Medicine, 3:, 93-102, DOI: 10.2147/MNM.S92187

2.Bone tissue overview

Line 128….” In the case of osteocytes, expression of single MMPs (MMP-2 and MMP-8) has been confirmed only in rat cells [34].  There is an increasing number of experimental and clinical  studies showing  that, beside MMP-2 and MMP-8,  osteocytes  may also  express several  MMPs including MMP-9, MMP-13,MMP-14 (also known as MT1-MMP) , MMP-16 and TIMPs as well. These proteinases appear  to play  a crucial role, in modulating various  specific steps  of bone tissue remodeling in physiological and pathological conditions.

 3.The role of metalloproteinases in physiology and pathology with special emphasis  on bone tissue

 Ad mentioned above, the activity of MMPs  is   regulated by TIMPs.   A physiological bone remodeling process  requires a  correct balance between MMPs and TIMPs. An  Imbalanced MMPs/TIMPs activity underlies major bone pathologies. Moreover, TIMPs may also  act as signaling molecules with cytokine-like activities, thus influencing various biological processes, including cell growth, apoptosis, differentiation, angiogenesis, and oncogenesis  (See  comments point 2 above).  Furthermore,normal and/or pathological bone remodeling may not be solely defined by the balance/imbalance between MMPs and TIMPs.  Emerging studies highlight the fact that MMs performance is also modulated  by f RECK  a membrane-anchored glycoprotein negatively regulating the activities of MMPs, involved in breakdown of the extracellular matrix (ECM) with tissue-specific and cell-anchored inhibitors increasing number of  specific miRNAs. 

4.1. Degenerative spine disease

An imbalance between TIMPs and MMPs plays an important role in intervertebral disc degeneration.  Recent studies  have shown that,  there is a significant difference  in the release of TIMPs and  specific MMPs, by tissues from patients with asymptomatic-IVDD compared to symptomatic –IVDD. These findings imply that TIMPs a nd MMPs  may have a  clinical role as biomarkers in distinguis between A-IVD and S-IVD patients  may help to improve the  diagnostic, prognostic, and therapeutic approach  to these patientsIntervertebral disc degeneration: symptomatic and asymptomatic distinguishers. Moreover, emerging studies highlight the fact that MMP performance is also modulated  by an increasing number of  specific miRNAs.

4.2. Malignant diseases of bone and dental cancers

 The  same considerations   as above can be made regarding these pathological conditions

( See comments point 1-3)

Discussion and Conclusion

Authors must add  an extensive version of these sections where they can appropriately  discuss  and  evaluate the  recent advances   on the  role of MMPs in  normal bone remodeling and some bone diseases. An overcrowded   “Summary Table”  cannot adequately  reflect the critical  Authors opinions  and, besides, is not  consistent with the editorial style of the journal

References

References  related to the topics  need to be revised and updated

Author Response

Dear Reviewer,

We would like to thank you very much for your careful review of our paper, entitled. "Importance of metalloproteinase enzymes group in selected skeletal system diseases" and for your accurate and useful suggestions. We have highlighted responses to suggestions in blue italics while changes to the manuscript have been highlighted in green. We hope that the corrections made will prove satisfactory and allow publication of our work in the “International Journal of Molecular Sciences”.

General comments

 The review of Kulesza et al. discuss the recent advances on the role of matrix metalloproteinases in normal and pathological bone remodeling and their potential clinical usefulness as biomarkers of some specific bone disorders.

Although the topic appears timely, the revision process of the manuscript has highlighted some critical issues and several flaws that the authors need to be addressed. In particular:

  1. In the Abstract, the Authors claim that the purpose of this review is to provide an overview on the role of MMPs in bone physiology and pathophysiology and in specific skeletal disorders. (lines 22-24.) On the other hand, in the Introduction, the Authors assert that the aim of the review is to provide insight into the most recent advances on the role of MMPs in the skeletal system and in the most common bone diseases, and to evaluate their clinical use as possible biomarkers of these pathological conditions. Apart from these divergent statements, because the aim reported in the Abstract appears to be consistent with the topic of the review, I would ask the authors to explain the reason why they have taken into consideration just some specific pathological conditions. Given the  epidemiologic,  and socio-economic burden  of  many other most commom malignant and non-malignant bone disorders it should be more fruitfull to provide insight into the most recent advances on the role of MMPs/TIMPs system and related microRNAs also in   non-malignant skeletal disorders such as primary and secondary osteoporosis, osteoarthritis (OA), rheumatoid arthritis (RA), Paget’s disease of bone (PDB), osteomyelitis and  cancer related bone disease  such as Multiple Myeloma (MM) and Metastastic Bone Disease (MBD). In this setting, emerging evidence highlights the fact that, in MM and MBD the MPPS/TIMPs/microRNA system appear to play a pivotal role in promoting these pathological conditions and are actually considered potential therapeutic targets and useful biomarkers that may foster the development of nover and more effective clinical treatment of these diseases.

Thank you for your pertinent commentary and for drawing attention to important issues related to this work. The disease entities selected for our work were chosen based on the pathophysiology of the above selected conditions. We have chosen to discuss only disease entities that are exclusively associated with changes in bone tissue. At the stage of preparing the manuscript, we took into account the disease entities mentioned by the reviewer, however, we decided not to include them for certain reasons.

First, osteoporosis and osteopenia were described extensively in our manuscript. The relevant passage describing these pathologies is included in lines 426-550. In addition, these disease entities are included in a summary table (Table 1. Summary of role and diagnostic potential of MMPs in selected bone diseases.).

Second, rheumatoid arthritis (RA). We initially planned to include this disease entity in the manuscript. However, given the fact that it is a systemic disease that leads to damage to various structures of the body, not just the skeletal system, we dropped the description of this disease, since our work, as mentioned, focuses on bone tissue. At the same time, we realize that this disease is associated with an increased risk of osteoporosis and osteopenia, however, there are currently no experimental papers that clearly show the association of MMPs with osteoporosis or osteopenia associated with RA.

Third, osteoarthritis was included in our study on the basis of osteoarthritis of the spine. This is because the vast majority of studies on the role of MMPs in osteoarthritis relate specifically to the spine. The amount of data on the role of these enzymes in terms of other types of osteoarthritis is very limited, so we focused only on spinal lesions in our study.

Fourth, we planned to include Paget's disease of bone in the paper as well, but there are currently no scientific reports on the role of MMPs in this disease entity.

Fifth, we had also planned to include multiple myeloma in our paper, however, given that it is a condition that damages bone tissue secondarily, we dropped the description of this condition. However, we are aware of the broad role of MMPs in this condition which includes osteolytic destruction of bone and facilitation of infiltration of myeloma cells into bone. In the future, we plan to publish a paper exclusively on the role of MMPs in multiple myeloma; the idea for such a publication was derived from the preparation of this manuscript. 

Sixth, we completely agree with the reviewer that our paper should include a chapter on metastatic bone disease. The manuscript has been completed with a corresponding chapter, which is contained in lines 519-555, and summarizes the existing knowledge on the role of MMPs in metastatic bone disease.

Seventh, we are aware of the role of the MMPs-TIMPs-miRNA system in the promotion of various pathological conditions, including in bone, but we will not include it in the paper because the manuscript deals exclusively with MMPs - their role in the physiology and pathology of bone and also considers their usefulness as potential markers.

  1. Regarding the role of MMPs in bone physiology and pathology, to be thorough a section describing the role of specific inhibitors of these enzymes, namely tissue inhibitors of metalloproteinases (TIMPs) in normal and pathological bone remodeling should be added. As the activity of MMPs is also regulated by TIMPs, a physiological bone remodeling process requires a correct balance between MMPs and TIMPs. On the other hand, a growing number of preclinical and clinical studies has provided evidence that. Imbalanced MMPs/TIMPs activity underlies major bone pathologies. Furthermore, TIMPs act as signaling molecules endowed with cytokine-like activities, thus influencing, besides normal and pathological bone turnover, various biological processes, including cell growth, apoptosis, differentiation, angiogenesis, and oncogenesis.

Thank you for your pertinent comment, we agree with the reviewer that tissue inhibitors of metalloproteinases (TIMPs) play an extremely important role in the function of MMPs and are involved in maintaining normal homeostasis of the body. However, we are not able to include these compounds as thoroughly in our manuscript. This is due to the fact that this manuscript focuses solely on MMPs, and adding TIMP-related issues would overextend the work. We realize that we have not nailed down the topic of TIMPs in the manuscript, but we have made the appropriate revisions in such a way as to familiarize the potential reader with this issue. In the future, however, we plan another review article that will address the role of TIMPs in the skeletal system.

Major Points

1.Introduction

Line 45 “….The exact role of MMPs in bone is not very well studied”…..To date, the involvement of MMPs in bone remodeling has been better defined following the experimental  use of MMP-knockout mouse models. These studies, by highlighting  a variety of bone abnormalities, have  provided clear evidence on the modulting  role of MMPs  in multiple stages of physiological and pathological bone remodeling processes (See for instance Hardy E et al.. Front Physiol. 2020 Feb 5;11:47. doi: 10.3389/fphys.2020.00047; Khoswanto C. J Oral Biol Craniofac Res. 2023 Sep-Oct;13(5):539-543. doi: 10.1016/j.jobcr.2023.06.002.  Liang HPO, (2016) Metalloproteinases in Medicine, 3:, 93-102, DOI: 10.2147/MNM.S92187

Again, thank you for your accurate comment, we realize that we expressed ourselves in an inaccurate manner. In the paper, we wanted to emphasize that the role of MMPs in the human skeletal system has not been well studied. In our work, we focused only on human models and did not consider studies on animal models, as this was not the intention of our manuscript. The sentence in line 45 has been corrected accordingly.

2.Bone tissue overview

Line 128….” In the case of osteocytes, expression of single MMPs (MMP-2 and MMP-8) has been confirmed only in rat cells [34].  There is an increasing number of experimental and clinical studies showing that, beside MMP-2 and MMP-8, osteocytes may also express several MMPs including MMP-9, MMP-13, MMP-14 (also known as MT1-MMP), MMP-16 and TIMPs as well. These proteinases appear to play a crucial role, in modulating various specific steps of bone tissue remodeling in physiological and pathological conditions.

Thank you for the pertinent comment. As we mentioned in our earlier responses, we have added a relevant paragraph to the manuscript we have added a relevant paragraph briefly describing TIMPs however, in this paper we are unable to add more data related to the present issue. We realize that the role of TIMPs in bone physiology and pathology is important, however, in our work we have focused only on MMPs. The addition of relevant chapters about TIMPs, would have significantly lengthened the work and made it too detailed which, in our opinion our opinion, would not be beneficial for the reception by the potential reader. As mentioned, we plan to publish another paper, which will be the next item in our series of reports on the role of the MMP-TIMP-miRNA axis in the system of skeletal system.

4.1. Degenerative spine disease

An imbalance between TIMPs and MMPs plays an important role in intervertebral disc degeneration.  Recent studies have shown that, there is a significant difference in the release of TIMPs and specific MMPs, by tissues from patients with asymptomatic-IVDD compared to symptomatic –IVDD. These findings imply that TIMPs and MMPs may have a clinical role as biomarkers in distinguish between A-IVD and S-IVD patients may help to improve the diagnostic, prognostic, and therapeutic approach to these patients.  Intervertebral disc degeneration: symptomatic and asymptomatic distinguishers. Moreover, emerging studies highlight the fact that MMP performance is also modulated by an increasing number of specific miRNAs.

We thank you for your apt commentary and sound analysis of the chapter. During the preparation of the manuscript, we also noted that scientific studies indicate an important role of TIMPs in the development of bone pathology. However, as we mentioned earlier, our work only considers the role of MMPs in selected bone diseases, and we do not plan to expand the manuscript to also include the role of TIMPs and miRNAs. However, as we mentioned earlier, given the large number of scientific reports, we plan to publish a similar paper on the role of TIMPs and miRNAs only.

4.2. Malignant diseases of bone and dental cancers

 The same considerations   as above can be made regarding these pathological conditions

(See comments point 1-3)

The role of TIMPs and miRNAs as prognostic and diagnostic markers in cancer has been scientifically validated for many years. These studies have been conducted in patients with breast, ovarian or colorectal cancer, among others. Some studies also address the role of these compounds in bone tumors. As in the case of the other conditions we described, we did not include the role of TIMPs because our work focuses exclusively on the role of MMPs.

Discussion and Conclusion

Authors must add an extensive version of these sections where they can appropriately discuss and evaluate the recent advances   on the role of MMPs in normal bone remodeling and some bone diseases. An overcrowded “Summary Table” cannot adequately reflect the critical Authors opinions and, besides, is not consistent with the editorial style of the journal.

Thank you for your comment. However, we cannot agree with the reviewer's suggestion to discuss and evaluate extensively the relevant sections of our work. Adding such a description would have the effect of making the work considerably longer and, as a result, would make it difficult for a potential reader to read. Furthermore, we do not believe that the summary table is overcrowded. We realize that it is extensive, but this is solely due to the large amount of research that our team analyzed. At the same time, it does not reflect our opinions, as it only serves to collect and clearly summarize the information that we described in the text. At the same time, the table will be corrected according to the editorial style of the journal.

References

References related to the topics need to be revised and updated.

Thank you for your comment, the literature will be reviewed and updated.

In conclusion, we would again like to thank you for your honest and conscientious review of our work. We hope that you will be fully satisfied with the changes made. If further concerns arise, we are open to conversation.

Sincerely,

Prof. Sławomir Ławicki

Reviewer 2 Report

Comments and Suggestions for Authors

The authors state the importance of MMP group with view of physiology and pathology in skeletal system diseases. They addressed new insights on MMP function. The available information gives an interesting study and suggests possible mechanisms for skeletal disorder. I have formulated below a few points that could help to improve the clarity of some information.   

1.     In introduction, line 50-53, rephrase the sentence for intended meaning.

2.     Line 67-69, rephrase the sentence.

3.     In figure 1, please show brief description. Check the meaning of heavy and light side.

4.     In table, indicate in which tissues these observations were made. I recommend authors should clarify this point. In addition, experimental system, tissue, or cells should be mentioned in the whole text.

5.     The same statement is provided on lines 120 and 147. Avoid repetitions and clarify the specific point.

6.     In section 3, reorganize or divide several section with view of each type of MMP.

7.     In section 4, reorganize this section. I recommend the organized description of MMPs.

Author Response

Dear Reviewer,

We would like to thank you very much for your careful review of our paper, entitled. "Importance of metalloproteinase enzymes group in selected skeletal system diseases" and for your accurate and useful suggestions. We have highlighted responses to suggestions in blue italics while changes to the manuscript have been highlighted in green. We hope that the corrections made will prove satisfactory and allow publication of our work in the “International Journal of Molecular Sciences”.

The authors state the importance of MMP group with view of physiology and pathology in skeletal system diseases. They addressed new insights on MMP function. The available information gives an interesting study and suggests possible mechanisms for skeletal disorder. I have formulated below a few points that could help to improve the clarity of some information.  

We would like to thank you for such a complimentary review of our manuscript and for your honest review and for pointing out corrections.

  1. In introduction, line 50-53, rephrase the sentence for intended meaning.

Thank you for pointing out the error, the work has been corrected accordingly. Any corrections have been highlighted in green. 

  1. Line 67-69, rephrase the sentence.

Thank you for pointing out the mistake. The sentence has been rephrased accordingly.

  1. In figure 1, please show brief description. Check the meaning of heavy and light side

Thank you for pointing out the mistake. Figure 1 has been removed from the manuscript. After the corrections were applied to the paper, this figure became redundant.

  1. In table, indicate in which tissues these observations were made. I recommend authors should clarify this point. In addition, experimental system, tissue, or cells should be mentioned in the whole text.

Thank you for your pertinent comment, the type of tissue studied is listed in the table - most often the studies involved expression in bone or tumor tissue and serum or plasma concentrations. All of the papers we entered were for work on a human model only; we did not include experimental work on animal models and in vitro, as this was not the intent of our manuscript. However, we believe that not including the above information will not adversely affect the merits of our manuscript.

  1. The same statement is provided on lines 120 and 147. Avoid repetitions and clarify the specific point.

Thank you for your comment. The phrase has been corrected accordingly.

  1. In section 3, reorganize or divide several section with view of each type of MMP.

Thank you for the pertinent comment. At the stage of preparing the manuscript, we planned to organize section 3 so that each MMPs is separated. Unfortunately, the amount of scientific data on MMPs is too small to separate them into the required sections. Nevertheless, we believe that Section 3 is presented in a readable way and that not dividing it into subsections will not affect the reception of the paper.

  1. In section 4, reorganize this section. I recommend the organized description of MMPs.

Thank you for your rightful comment. At the planning stage of the manuscript, also in this section we planned to divide the paragraphs into subsections taking into account individual MMPs. However, we felt that performing such a procedure resulted in repeated repetition of the same information in the text, which translates into a significant deterioration of its quality. Therefore, an orderly description of the role of individual MMPs was presented in a summary table which relates strictly to the text. We believe that this is the way our work will be most accessible to the potential reader.

In conclusion, we would again like to thank you for your honest and conscientious review of our work. We hope that you will be fully satisfied with the changes made. If further concerns arise, we are open to conversation.

Sincerely,

Prof. Sławomir Ławicki

Round 2

Reviewer 2 Report

Comments and Suggestions for Authors

The manuscript is improved adequately. I recommend the abbreviation section should be included below the table.